# Exploring the Role of Community Exercise Rehabilitation Centers through the Rehabilitation Experiences of Patients with Musculoskeletal Disorders: A Qualitative Study

**DOI:** 10.3390/healthcare12010092

**Published:** 2023-12-30

**Authors:** Dongjoo Park, Jiyoun Kim

**Affiliations:** 1Institute of Human Convergence Health Science, Gachon University, Incheon 21936, Republic of Korea; ehdwn0201@hanmail.net; 2Department of Exercise Rehabilitation, Gachon University, Incheon 21936, Republic of Korea

**Keywords:** musculoskeletal disorder, exercise rehabilitation system, living lab, local community, older adult, chronic pain

## Abstract

We investigated the perception of community sport rehabilitation centers by examining the experiences of living lab participants and discussing the centers’ roles. From 50 living lab participants in the exercise rehabilitation center, in-depth interviews were conducted with the 10 among them (three males and seven females) who had high participation rates and consented voluntarily. The data collected through participant observation and a review of the literature were analyzed through inductive categorization. The findings show the points as follows: (a) owing to inadequate cognitive and physical accessibility, the older population faced challenges in utilizing the center, causing deficiency in the requisite information and knowledge essential for effective rehabilitation; (b) human and material services, including older adult rehabilitation instructors, systematic management, and service platforms were provided to the participants through the centers; (c) after 12-week program operation, participants experienced improved physical health, and by engaging in the rehabilitation exercise services, developed positive perception of the center. Participants desired to continue with the program and were willing to recommend it to others. These findings emphasize the importance of specialized instructors to older-adults’ physical activities, implementing systematic data management and utilization practices, and the collaboration between healthcare institutions and local communities. This is particularly important because of the rapid rise in the older adult population.

## 1. Introduction

Older adults are exposed to various types of chronic pain and the most common of these is musculoskeletal chronic pain, which is experienced by approximately 81% of the older adult population [1]. Specifically, most older adults experience discomfort attributable to chronic knee pain stemming from factors such as muscular weakness, weight-bearing challenges, and joint degeneration. The resulting knee dysfunction, induced through chronic pain in this demographic, precipitates gait and activity impairments leading to diminished psychological well-being and a compromised quality of life. Notably, there is a high propensity for the recurrence of knee pain even after pain alleviation through therapeutic interventions. To manage knee pain in older adults, steady muscle exercises and rehabilitation are the most effective treatment methods [2].

Ferretti et al. [3] observed that older individuals with sedentary lifestyles exhibited higher musculoskeletal pain intensity than their physically active counterparts, emphasizing the necessity for targeted exercise interventions to alleviate such pain. Furthermore, Hirase et al. [4] demonstrated that an intervention program promoting increased physical activity could prevent musculoskeletal pain and effectively enhance cognitive function, elevate physical activity levels, and improve psychological well-being. Expert-mediated physical activity programs have emerged as potent strategies for pain prevention among older adults. Additionally, numerous previous investigations have underscored the efficacy of exercise rehabilitation programs in preventing pain and fostering functional improvement in older individuals experiencing knee pain [5,6,7].

In alignment with the findings of these preceding studies, expert-led appropriate exercise programs for older individuals with knee ailments contribute to disease management and enhance daily life satisfaction. However, in domestic hospitals, rehabilitation is provided only to patients with acute illnesses. To manage the disease after discharge, the skills and experiences acquired in the hospital should be linked and reflected in the community exercise rehabilitation center. However, because there is no continuous exercise rehabilitation system, most patients discontinue functional rehabilitation. Therefore, older adults with musculoskeletal disorders should use private facilities [8].

The exercise rehabilitation center is situated in the local community and serves the function of preventing ailments and facilitating the safe return of individuals to the community by enhancing physical, psychological, and social well-being. Specifically, the “senior living lab” provides comprehensive health services by implementing exercise programs aimed at preventing and facilitating recovery from ailments in the older adult population. Furthermore, it functions to enhance awareness of overall health and improve quality of life [9].

According to the National Life Sports Survey reported by the Ministry of Culture, Sports, and Tourism, Republic of Korea, in 2022, 12.8% of the respondents said they received exercise prescriptions or counseling services based on regular measurements or examinations. After surveying the reasons for not using the service, 43.1% of the respondents said they had no information or knowledge about the management method. Among them, 49.6% were in their 60s, 55.9% were in their 70s, 75.4% used private facilities, and 24.6% used facilities operated by national and local governments. Among them, 93.7% of those in their 20s, 91.7% in their 30s, 86.9% in their 40s, and 78.5% in their 50s used private facilities. However, 39.3% of those in their 60s and 17.2% in their 70s or older showed a large difference from those in their 20s and 50s [10]. Most age groups with musculoskeletal disorders do not have exercise prescriptions for regular measurements or screening. In particular, it was reported that older adults in their 60s and 70s lacked information and knowledge about physical fitness management methods and the management of musculoskeletal diseases.

Older adults with musculoskeletal disorders would benefit from a better understanding of the physical and mental changes associated with the disease and the human-led services that exchange emotions with them are important [11]. However, older adults in Korea currently have low awareness of community sport rehabilitation centers; therefore, they visit fitness centers or manage diseases through online media without distinguishing between sports for all and exercise rehabilitation [12]. The environment of the fitness center and the human-led service of the leader are operated for people in their 20s and 30s; therefore, it is difficult for older adults with musculoskeletal disorders to receive exercise rehabilitation services. In addition, self-treatment through online media is less efficient for management because of the difficulty in performing operations and the provision of unclear information [13]. This erroneous management method slows the time of returning to daily life due to recurrent pain and deteriorating physical function and repeats a vicious cycle of returning to the hospital [14]. In other words, the absence of an exercise rehabilitation system after discharge leads to secondary damage and hospital revisits, resulting in increased medical expenditure.

Meanwhile, in South Korea, to establish and develop exercise rehabilitation services linked to the local community after discharge, the Ministry of Health and Welfare, the Ministry of Science and Information and Communication Technology (ICT), and the Ministry of Culture, Sports, and Tourism collaborated in 2021 to initiate national research and development (R&D) for the development of public exercise rehabilitation program services. The study developed the exercise rehabilitation functional code (ERFC), a systematic code that provides comprehensive therapeutic information for patients requiring community exercise rehabilitation, especially those who have completed treatment at medical institutions.

The evaluative components of the “ERFC” included gait and joint range of motion, balance, pain, muscle strength, and cardiovascular endurance. Based on these measurements, the results are systematically managed through an “ICT-based exercise rehabilitation service platform” implemented with ICT. Subsequently, tailored exercise programs were prescribed by instructors according to individual characteristics. The effectiveness of these programs was verified over a three- to six-month implementation period (Ministry of Culture, Sports, and Tourism R&D Program No. SR202104001). Therefore, we aimed to explore the experiences of participants involved in national R&D to establish a comprehensive, long-term, public exercise rehabilitation program. By examining these participants’ experiences, this study sought to gain insights into their perceptions of local community exercise rehabilitation centers. Additionally, this research aimed to engage in an in-depth analysis of participants’ experiences in exercise rehabilitation programs to facilitate a comprehensive discussion of the role of local community exercise rehabilitation centers.

## 2. Materials and Methods

### 2.1. Data

#### 2.1.1. Research Participants

To achieve this study’s objectives, participants involved in national R&D were selected to establish a comprehensive, long-term, public exercise rehabilitation program. The criteria for participant selection were divided into two phases.

For the first phase, 30 individuals were chosen from an initial pool of 50 participants who actively engaged in a 12-week exercise program as part of a senior living lab for R&D. In the second stage, the final 10 participants (three males and seven females) were selected according to the result saturation criterion among those who were judged to provide sufficient opinion and qualitative data for in-depth interviews. The age of these final 10 individuals ranged from 60 to 70 years. These participants were experiencing conditions such as knee osteoarthritis, knee joint disorders, knee joint pain, and other musculoskeletal-related knee ailments. The basic characteristics of the participants are presented in Table 1.

#### 2.1.2. Living Lab

To comprehensively analyze participants’ experiences in the exercise rehabilitation program, an understanding of the living lab’s operational process is essential. The living lab’s operational process involved recruiting male and female individuals aged 50–70 years who experienced discomfort in daily life due to knee conditions (degenerative joint arthritis and meniscus damage). The recruited participants received the Korean standard classification of disease and causes of death (KCD) code based on a specialist’s diagnosis. Upon visiting the living lab, they completed a research participation consent form, provided personal information, and filled out health screening questionnaires (health screening) as well as readiness for physical activity surveys. This study was approved by the institutional review board of Gachon University (No. 2023-018, 22 March 2023). Before starting the exercise program, ERFC assessment elements, including pain index, joint range of motion, flexibility, cardiovascular endurance, muscle strength, muscle endurance, balance, coordination, gait, and balance, were evaluated. The measured assessments were entered into the “ICT-based exercise rehabilitation service platform” to enable the instructor to prescribe exercise programs tailored to individual characteristics. Participants participated in a prescribed exercise program for 12 weeks. Additionally, to systematically manage exercise during its progression, interim measurements (balance, muscle strength, and muscle endurance) were conducted at two-week intervals. Based on the interim measurement results, the form, intensity, and frequency of exercise were adjusted and exercise was prescribed accordingly. Furthermore, to ensure systematic management, the “ICT-based exercise rehabilitation service platform”, implemented with ICT, was utilized to manage and monitor community patients. The operational process of the living lab is shown in Figure 1.

### 2.2. Data Collection

In addition to participant observation, in-depth interviews were conducted with the research participants to collect qualitative data addressing the research questions through detailed content and genuine conversations. In addition to participant observation and in-depth interviews, the literature, online resources, and newspaper articles were used as additional sources of information for the study.

#### 2.2.1. Participant Observation

To conduct participant observations, we obtained consent from the exercise center staff and participants involved in the established living lab. Subsequently, from March to August 2023, we visited the living lab at least twice a week and documented the activities observed in research notes.

#### 2.2.2. In-Depth Interview

We collected reliable qualitative data through in-depth interviews and made efforts to visit the living lab 1–2 times per week to establish a rapport with the research participants. In-depth interviews were conducted after establishing mutual trust and the interview location and timing were arranged through prior agreement to accommodate the participants’ exercise schedules, leading to visits to the living lab. Recordings were conducted with the participants’ consent, which was obtained before the interviews, and each session lasted between 40 min to 1 h. To ensure the quality of the questions, a semi-structured interview (see Appendix A) script was developed and validated by Park, DJ. and Kim, J. This process aimed to align the questions with the central objectives of the study and ensure clarity in language and relevance to the research domains [15]. The format of the in-depth interviews involved questions posed in two stages comprising primary interview questions and secondary follow-up questions. During the primary interview, comfortable conversations were used to gather information about the participants’ life backgrounds, current experiences, and various aspects related to the research topic. Once a comfortable atmosphere was established, the second round of interviews gradually delved into more in-depth discussions by introducing questions directly related to the research topic. To assess awareness on community exercise rehabilitation centers and deeply explore the participants’ experiences of exercise rehabilitation programs, the following questions were posed.

#### 2.2.3. Auxiliary Research Data

The supplementary research materials primarily included content from relevant books, newspaper articles, and magazines. Additionally, various research materials from institutions related to older adult health management (such as the Korea Health Management Association, Korea Sports Promotion Foundation, and Statistics Korea) were comprehensively reviewed. This encompassed research data, website content, articles, and the literature on older adult health. These supplementary sources were used to enhance the overall research perspective.

### 2.3. Data Analysis

The recorded in-depth interview data were transcribed into textual data using a computer. The transcribed data were analyzed based on grounded theory by Park, D. and Kim, J. and one holder of a Ph.D. in the sociology of sport. No specific qualitative analysis program was utilized in this study. The analysis was conducted manually through methods such as thematic coding and content analysis. In the first open coding stage, the data were systematically organized by categorizing and coding repeated words, subthemes, and themes. Each theme was assigned conceptual titles and definitions for each concept were established to facilitate a systematic structure for the categorized content. In the second axial coding stage, the researchers explored the relationships between the themes derived from open coding. They underwent a process of determining importance, prioritization, and grading.

The process of identifying and hierarchically organizing relationships between themes was conducted based on the following criteria of (1) mutual exclusion: each interview unit was assigned to only one category; (2) exhaustiveness: the entire text was analyzed; (3) homogeneity: the analysis of similar categories followed consistent principles; (4) objectivity: different parts of similar data were encoded in the same way; and (5) validity: categories were considered valid if they aligned with a suitable structure or framework for achieving the intended purpose.

### 2.4. Truthfulness of the Research

The methods outlined by Denzin and Lincoln [16], such as the triangulation technique, long-term and repetitive observations, the utilization of reference materials, and advice and feedback from peer researchers were employed to enhance the truthfulness of the research.

## 3. Results and Discussion

### 3.1. Older Adults Wandering from Hospital to Hospital as Rehabilitation Refugees

The current health insurance fees for hospitals, including tertiary general hospitals, apply a perception-based hospitalization fee for stays lasting more than 16 days making prolonged hospitalization challenging without specific reasons, such as surgery or accompanying conditions [17]. Consequently, most patients with musculoskeletal disorders return to the community with insufficient preparation and those requiring functional recovery seek treatment at multiple hospitals. Society is becoming increasingly aware of this phenomenon to the extent that the term “rehabilitation refugees” has emerged to describe these patients’ situations [18].

#### 3.1.1. Difficulty of Post-Discharge Management

The prevalence of knee joint disorders among older adults in modern society is increasing. As the condition progresses, joint stiffness, reduced range of motion, and pain lead to limitations in mobility and walking [19]. The resulting functional decline and activity restrictions due to pain contribute to a sense of role loss and fear, negatively affecting emotional well-being. Although the participants acknowledged these phenomena, they expressed a sense of helplessness, stating that apart from visiting various hospitals to alleviate their pain, there seemed to be little else they could do.


*“I was a little bit rehabilitated at the hospital where I had surgery. I was discharged from the hospital for about two weeks and I was treated at the hospital and that’s all. I do not know how to go to the hospital and manage it.”*
(Participant H)


*“I was rehabilitated for a while before I was discharged from the hospital. I was worried about obtaining the same health as before. I did not do anything except go to the hospital.”*
(Participant B)


*“In the hospital, I walked a little because I often walked without difficulty, but I could not just walk about vaguely, so I did not exercise while I was exercising. I went to another hospital, but…”*
(Participant D)

The characteristics of surgery for knee joint disorders include an abnormal joint range of motion after the procedure necessitating an increase in mobility through rehabilitation. Appropriate rehabilitation intervention leads to pain reduction, increased joint range of motion, enabling walking, and, consequently, improved overall satisfaction with daily life [20]. However, the participants in this study reported relying more on hospital visits for temporary pain relief than on achieving sustained recovery through exercise rehabilitation. Participant H only received rehabilitation therapy during the hospital stay after surgery and managed pain by visiting the hospital after discharge. Additionally, participants B and D engaged in brief exercises but lacked proper management knowledge and relied solely on outpatient treatments at the hospital. This aligns with a 2022 report from the Ministry of Culture, Sports, and Tourism that indicated that 90.2% of those in their 60s and 90.5% of those aged 70 and above responded that they did not use exercise prescriptions or counseling services. The most common reason for not utilizing these services was cited as “lack of information or knowledge about musculoskeletal management methods”, which supports the findings of this study.

#### 3.1.2. Accessibility Concerns

Older individuals who do not have access to systematic exercise rehabilitation programs manage their health by visiting fitness centers or utilizing online media for exercise programs. However, services provided by local fitness centers are generally tailored for adults, making it challenging to effectively manage musculoskeletal disorders in older adults. Moreover, online exercise programs suffer from unclear information and difficulty in executing movements, making it challenging to engage in exercises without professional guidance [12]. To address these issues, specialized health management centers operating within local communities are needed. South Korea has an institution called the national fitness award that provides rehabilitation services. Designated as a nationally certified institution, it scientifically measures and evaluates physical fitness and offers sports welfare services by providing exercise counseling and prescriptions [21]. However, the fitness measurement methods and criteria presented by the national fitness award are based on standards for the general population rather than patients, making it difficult for older individuals with knee disorders to utilize these services. Therefore, older individuals in need of exercise rehabilitation should consider utilizing community-based rehabilitation centers specializing in musculoskeletal health. However, many older adults express reluctance towards the fitness center environments and services provided by instructors.


*“There is a burden for people like us to go. I would go and exercise, but among the young people who are loud and heavy I thought it would be a nuisance if an uncle like me went.”*
(Participant C)


*“I went to a fitness center near my house because I wanted to exercise. However, I did not ride my bicycle a few times because it was difficult to use the equipment. Rather, it seemed to be a hindrance to my young friends when I was using it.”*
(Participant E)


*“I went to the gym where my son went to the gym for a while, but I did not go for a long time. It was dark and dizzy. But, we do not know the exercise. I just rode a treadmill and rode a bicycle.”*
(Participant F)

To effectively guide the physical activity of older adults, a more detailed approach is required to address both physical and emotional changes. Specialized knowledge of musculoskeletal diseases in older adults is necessary and the process of exchanging emotions is crucial [11]. Furthermore, the space in which older adults engage in physical activity goes beyond a simple physical environment, offering opportunities for various social relationships, role expansion, and establishing a social identity in old age. In other words, meaning is attributed to the space where older adult exercise is significant. Therefore, personalized services provided by experts and exercise environments play crucial roles in the management of musculoskeletal diseases in the older adult population. However, recent trends, particularly among individuals in their 20s, such as the popularity of body profiles for self-expression have led fitness and public exercise centers to operate interiors and programs tailored for the general adult population [22]. This shift may have made it challenging for older adults and those with musculoskeletal disorders to visit exercise centers for functional recovery and health promotion. Participant C, for instance, expressed reluctance to visit exercise centers, referring to them as “noisy places”. Participants E and F, despite having experienced exercise centers, stopped participating because of a lack of understanding of the atmosphere and exercise methods in these centers.

### 3.2. Participants Changing to Systemic Management

Participants reported having past experiences with exercise centers for general health management, but none had experienced specialized exercise programs guided by professional instructors. Most participants were in a state where functional rehabilitation had been discontinued after discharge. Therefore, they had low awareness of the exercise programs provided by local community exercise rehabilitation centers before participating in this study and expressed concerns about their expected effectiveness.

#### 3.2.1. Secretary for Health

Health exercise managers receive requests from hospitals, health centers, fitness centers, etc., through the referral of physicians and guide and manage exercise accordingly [23]. In this study, the participating health exercise managers (instructors) recruited participants for the living lab, facilitated their visits to the hospital to obtain the Korean standard classification of disease and cause of death (KCD) codes, and assisted them in completing health screening questionnaires. They conducted evaluations using the systematic ERFC that provides comprehensive therapeutic information regarding existing patients. The measured assessments were input into the ‘ICT-based exercise rehabilitation service platform’, allowing instructors to prescribe personalized exercise programs based on individual characteristics. Additionally, they conducted mid-term measurements every two weeks to supplement the exercise prescriptions. This systematic management instilled high trust and satisfaction among the participants.


*“My teacher was a health secretary. I do not know how meticulously I take care of things, I always check my condition, pain, etc., and every time I measure it periodically and I explain in detail how I feel better and am given homework for going home and exercising.”*
(Participant J)


*“I thought I was doing well when I watched YouTube alone or did this or that, but I met my teacher this time and I realized how important it is to have a class with a professional teacher. It is a simple thing, but it has a good effect on my body and the pain disappears.”*
(Participant B)


*“He gave me homework so that I could go home and manage it. He contacted me halfway to check my health condition and meticulously managed me.”*
(Participant I)

Participant J referred to the instructor as a “health secretary”. The instructor ensured a thorough understanding of the participants’ health status before exercise, demonstrated accurate recovery capabilities through measurements, and provided tasks for at-home management after exercise. Participant B highlighted the difference between exercising alone on YouTube and engaging in professional exercises that lead to pain relief. Participant I expressed satisfaction with the health exercise manager’s meticulous management skills that ensured that health management was extended into daily life. Quigley et al. [24] stated that “the exercise prescription service environment (including health exercise managers’ guidance on exercise programs and personal services) provides scientific and specific judgments on healthy fitness. It is based on individualized exercise programs prescribed by experts to maximize the effects of fitness and health promotion.” The term “health secretary” used by the participants implies that instructors maximize the effectiveness of knee disease recovery through systematic management and personal services. This, in turn, forms a high level of trust and satisfaction among participants and health exercise managers.

#### 3.2.2. Efficient Management Using an ICT-Based Service Platform

In this study, an ICT-based exercise rehabilitation service platform developed to align with community-based exercise rehabilitation services was applied to participants for management purposes. The utilization of the exercise rehabilitation service platform involved the following points: entering participants’ personal information and health data (KCD codes, measurement records, etc.), prescribing exercises based on the entered participant information, inputting exercise performance results for systematic management, and conducting re-measurements at two-week intervals to supplement exercise prescriptions as well as how participants were provided with health information in a task format to check and perform exercises at home. Participants who experienced the ICT-based exercise rehabilitation service platform expressed high levels of satisfaction.


*“I thought it was a notepad that only inputs my information. I was using it to tell me about my steady knee condition. It was so amazing. I can see how my knee is getting better and what exercise I did today.”*
(Participant F)

“*I thought I could get care in this way. I was more confident when I saw my knee getting healthy. I checked my health condition with my cell phone and my daughter was doing the homework tests. My daughter exercised with me at home. My daughter was also satisfied*.” (Participant E)

“*It was so good at showing my knee condition on the graph. So, when I go home, my son always looks at the record. What exercise I did, how much I improved, it looks like an elementary school student notice*.” (Participant H)

Participant F expressed great satisfaction with using the ICT service platform, highlighting the ease of checking changes in knee condition through exercise performance and readily confirming the prescribed exercise program. Participant E found it convenient to quickly check health status on a phone and noted that family members were also satisfied with the ICT service platform as they could comfortably check and participate in exercises together at home. Participant H expressed great satisfaction with the ease of checking health status and sharing it with family members.

Historically, older adults have been marginalized in the adoption of ICT. This characterization has perpetuated a stereotype that identifies older adults as the age group with the lowest rates of computer and internet usage. Currently, older adults in South Korea are actively embracing the advancements in ICT and this progress significantly contributes to the promotion of independent living among them. In South Korea, the internet usage rate among individuals aged 60 years and over has increased from 30% in 2011 to 88.8% in 2018, with approximately half of older adult users participating in social networking services (SNSs) such as Twitter, Facebook, and blogs [25].

Oh and Jung [26] reported that “healthcare services for the medically vulnerable older adult not only enhance medical convenience and efficiency, resolving medical blind spots, but also address social and psychological weakening, realizing a higher quality of life through the formation of social networks.” Thus, using the ICT service platform for management fulfills the role of a health assistant and provides participants with high satisfaction, significantly aiding in exercise performance.

#### 3.2.3. Service Provided by the Exercise Rehabilitation Center

Consumers using a service tend to evaluate the overall service through the visible physical service environment, which is a crucial marketing strategy for service providers. Such environmental services are significant for passive older adults and patients exposed to new environments, making them crucial [27]. Participants acknowledged the challenges in using facilities within their age groups or considered it disruptive to younger friends. They initially felt a sense of unfamiliarity with the exercise center. However, after visiting the center before starting the exercise program and witnessing an environment different from what they had previously imagined, they expressed high satisfaction.


*“I felt like I was at a museum. The song was calm and the atmosphere was comfortable. The fitness center I went to the other day was dark, loud, and a little confusing, but the center here was calm and relaxed.”*
(Participant A)


*“I felt like I was going to a very neat hotel. The interior was so comfortable and there were places where older people like me could go.”*
(Participant F)

Participants who visited the exercise center all used terms like “hotel” and “art gallery”, expressing great satisfaction with the different appearance from what they had envisioned. Participant F mentioned a change in perception toward exercise centers, stating that there are places even for older individuals. This phenomenon can be explained through the term “servicescape” referring to the physical environment where service creation and delivery occur, distinct from natural or social environments [28]. Mody et al. [29] stated that “As consumers perceive higher servicescape quality, customer satisfaction increases. A well-designed physical service environment can increase consumer expectations and satisfaction.” In other words, participants experienced increased satisfaction through systematic management and environmental factors, such as a stable interior resembling high-quality art galleries or hotels, that differed from their prior expectations or experiences with exercise centers.

### 3.3. Active Older Adults with Healthy Knees

#### 3.3.1. A Healthy Life Recovered through Functional Recovery

Persistent pain caused by knee disorders significantly affects walking ability and has various consequences. In many cases, patients experience depression, unnecessary energy expenditure, reduced appetite, weight loss, and overall weakness, all contributing to a diminished quality of life [30]. Similarly, the participants in this study experienced significant discomfort in their daily lives due to pain before engaging in the exercise program. Activities such as going out and socializing with friends became challenging and the frequency of outings decreased. However, through systematic management, the participants experienced significant functional recovery allowing them to regain aspects of their previous daily lives.


*“I have a lot of stairs in front of my house, so I was stressed to go out once, but now I can do it. I feel comfortable using public transportation, so I can now do shopping alone at the mall in front of my house.”*
(Participant A)


*“Before I started working out, going to the hospital regularly was on my schedule. But now my first priority is going to the exercise center.”*
(Participant J)


*“The number of times I go to the hospital has decreased a lot. It is amazing. I started to work out and I was worried about what to do if I got sicker. I do not go to the hospital now.”*
(Participant D)

Participants described numerous changes in their daily lives. Participant A emphasized that the most significant change through exercise was the reduction of stress related to outings and the increased convenience of using public transportation improved the quality of daily life. Participant J noted that regular hospital visits were a major part of their routine, but now visiting the exercise center has become the primary routine. Participant D mentioned a decrease in hospital visits, indicating that exercising at the center during the time allocated for hospital visits contributed to increased confidence. The participants experienced substantial functional recovery through systematic management, improving their quality of daily life and significant restoration of self-esteem. This aligns with the findings of Yu and Liu [31] who reported that psychological intervention therapy reduced negative feelings and enhanced hope levels in older adult patients undergoing knee replacement surgery as pain reduction had a positive impact on emotions such as tension, depression, and fatigue. Similarly, a study by Lee et al. [32] on progressive dynamic balance training in older adult women who had underwent knee joint replacement surgery revealed that the training program contributed to the recovery of physical function and balance ability, enabling patients to perform basic household chores and light exercises and ultimately improving their quality of life. In summary, appropriate exercise rehabilitation interventions positively influence emotional states by reducing pain, increasing joint mobility, enhancing walking ability, and subsequently improving daily life functions in the older adult population.

#### 3.3.2. Relationship between Systematic Management and Re-Participation Decision Making

Positive exercise experiences significantly influence expectations of and personal satisfaction with physical activities. Satisfactory emotions derived from exercise serve as essential factors in maintaining long-term exercise adherence, acting as indicators of whether individuals will continue to participate in the present and future [33]. Participants who required exercise rehabilitation received systematic management after receiving medical treatment. Issuing the KCD codes and measurements based on the ERFC for exercise rehabilitation provides a means for transmitting therapeutic information between hospitals and local communities. Instructors prescribed and managed exercise programs tailored to individual goals, ensuring that the participants felt comfortable managing their health through the 12 weeks of continuous care through an ICT-based exercise rehabilitation service platform. These processes instilled high levels of satisfaction and trust in the participants, ultimately leading them to pay willingly for ongoing management and confirm their intention for sustained participation in the program.


*“Now that I know the taste of exercise, I will continue to participate. I have a desire to be healthier. And now I have a schedule, so I get busy and lively.”*
(Participant I)


*“I have already registered as a member of the program. I have to do it steadily now. I have already recommended it several times to my acquaintances. I also have a friend to exercise with me.”*
(Participant G)


*“Now, I have decided to go to exercise regularly. My family also wanted to go to the exercise center and get care. Now, if I am sick, it is more comfortable to be managed at the exercise center than at the hospital.”*
(Participant B)

Participant I expressed notable contentment with exercise-based management and independently decided to visit the exercise center. Additionally, Participant G enrolled in a exercise center membership for ongoing care. Participant B opted for exercise center management over hospital care for pain relief. The intention to re-engage at the fitness center can be conceptualized as the consumer’s commitment to continuing the utilization of exercise services in the future. Meeting participants’ expectations and demands emerges as a pivotal strategy for fostering loyalty and securing a competitive advantage [34,35].

Naim et al. [36] stated that “Consumers, when assured of service quality in selecting pharmaceuticals and health services, make purchases without hesitation. However, they often encounter confusion in the purchase process due to insufficient information”. Therefore, institutions delivering health services must disseminate ample information to participants and consumers through promotional efforts, highlighting service quality and type to cultivate trust.

In summary, participants initially harbored low trust in the exercise rehabilitation center prior to experiencing the living lab exercise program. However, their satisfaction was elevated through the implementation of systematic services. Furthermore, participants’ willingness to incur personal medical expenses after the exercise program’s conclusion implies a sustained engagement driven by increased trust through functional recovery.

## 4. Conclusions

Chronic knee pain in older adults leads to a decline in knee function, causing limitations in walking and overall activity. This leads to an increase in psychological distress, deterioration of the quality of daily life, and, despite pain relief through treatment, a significantly heightened likelihood of recurrence. To manage such knee conditions, a combination of pharmacological treatments and non-pharmacological interventions, including strength exercises and rehabilitation, is commonly employed. However, owing to the absence of a community-linked exercise rehabilitation system after discharge, most older adults experience a discontinuation of functional rehabilitation. Systematic management-deprived older adults often attempt self-health management through exercise centers or online media. However, by engaging in mostly inappropriate exercises or failing to control the intensity, they are often caught in a vicious cycle of delayed return to daily life and repeated hospital visits.

To address these issues at the national level, R&D efforts are underway to develop comprehensive public exercise rehabilitation program services. Through this effort, we developed the ERFC, a systematic framework for adequately conveying therapeutic information to existing patients. This has facilitated the connection between medical institutions and local communities, creating an environment where patients with a knee disorder can engage in precise and safe exercise rehabilitation. Thus, by examining the perception of local community exercise rehabilitation centers from the perspective of older adults, further development of community-linked services can occur. Therefore, this study aimed to explore the perception of local community exercise rehabilitation centers by examining the experiences of participants who were involved in national R&D to develop comprehensive public exercise rehabilitation programs. The researchers sought to analyze in-depth collected information and to discuss the role of local community exercise rehabilitation centers.

The results indicated a lack of a continuous exercise rehabilitation system connected to the local community, making it difficult for older adults to find effective methods to manage musculoskeletal disorders. Furthermore, mainstream exercise centers, including fitness centers that can be managed daily, were found to operate with interiors and services tailored for individuals in their 20s and 30s, posing difficulties for older adults to access.

Exercise instructors prescribed individualized exercise programs based on the KCD codes, health screening questionnaires, and the ERFC. They managed participants using an ICT-based exercise rehabilitation service platform. This combination of environmental services for older adults and exercise instructors’ expertise, including medical knowledge on diseases and ample experience in operating exercise programs, played a significant role. Through systematic management using data utilization and personalized services, participants’ trust in the exercise rehabilitation center increased.

The exercise rehabilitation service led to a reduction in knee pain and improvement in functionality. Consequently, the participants experienced increased awareness of exercise participation, active engagement in exercise practices, and elevated satisfaction with daily life. Ultimately, the improvement in participants’ satisfaction with their daily lives generated a high level of contentment and trust, as evidenced by their willingness to continue with the service by voluntarily paying for ongoing management.

It is imperative for instructors to prioritize systematic measurement over subjective judgments in exercise prescriptions, emphasizing the need to utilize data effectively. Especially because older adults may experience anxiety in new environments, it is crucial to strategically arrange facilities by modifying the equipment and environment to enable free and comfortable use of the facility. Additionally, fostering psychological well-being is essential in order to promote opportunities for social engagement. Additionally, ensuring the provision of personalized services by qualified instructors is crucial. Collectively, these aspects represent critical factors for enhancing the quality of community-based exercise rehabilitation services.

This study confirmed that low awareness of the environment and personnel services at community-based exercise rehabilitation centers negatively impacts the participation of exercise rehabilitation participants. Additionally, measuring the medical history and functional exercise data of exercise rehabilitation participants emerged as a crucial element for systematic exercise rehabilitation service management, actively influencing confidence in exercise participation and sustained engagement. In summary, by providing appropriate services through systematic management that is provided by instructors and by creating favorable environments and personnel services at exercise rehabilitation centers, the active participation of older adults can be promoted, thereby activating community-based exercise rehabilitation centers.

In 2021, the Ministry of Health and Welfare introduced “My Health Way” to connect medical institutions and local community data providers to facilitate active health management by the public. [37] The exercise rehabilitation services in this study enhanced systematic management by leveraging data and fostered a perception of trustworthy exercise rehabilitation services. Based on these findings, it is anticipated that future studies will provide sufficient evidence for implementing comprehensive community-based exercise rehabilitation programs integrated with hospitals. Consequently, we recommend that policy research be conducted in the future to further develop connected exercise rehabilitation services between hospitals and local communities.

## Figures and Tables

**Figure 1 healthcare-12-00092-f001:**
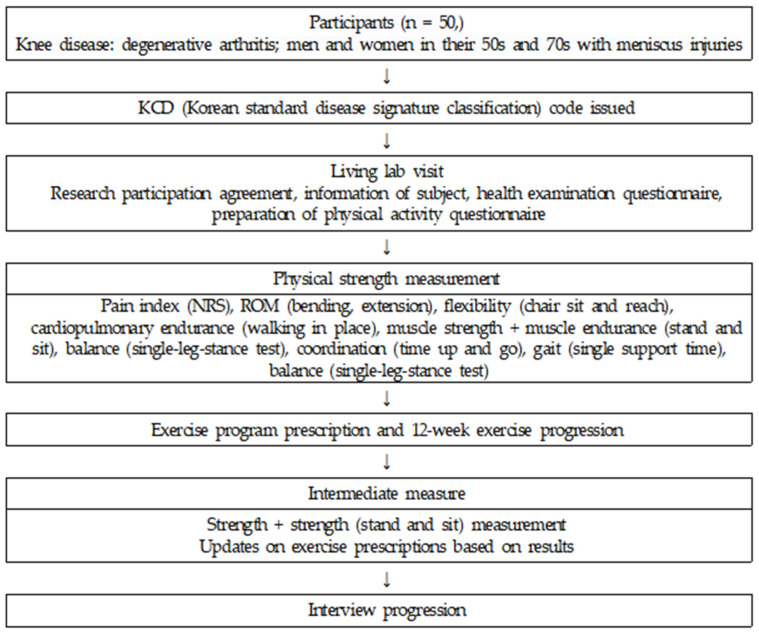
The operational process of the living lab.

**Table 1 healthcare-12-00092-t001:** Participants’ characteristics.

	Name	Sex	Age (Years)	Knee Condition
1	A	Female	72	Knee osteoarthritis
2	B	Female	60	Musculoskeletal-related knee ailments
3	C	Male	77	Knee osteoarthritis
4	D	Female	60	Musculoskeletal-related knee ailments
5	E	Female	64	Knee joint disorders
6	F	Male	64	Musculoskeletal-related knee ailments
7	G	Female	74	Knee joint pain
8	H	Female	77	Musculoskeletal-related knee ailments
9	I	Male	67	Knee joint pain
10	J	Female	74	Knee joint disorders

## Data Availability

Data are available upon request due to restrictions. The data presented in this study are available upon request from the corresponding author. The data are not publicly available.

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
