# Peer review of "Exploring the Role of Community Exercise Rehabilitation Centers through the Rehabilitation Experiences of Patients with Musculoskeletal Disorders: A Qualitative Study"

_healthcare, 2023, doi:10.3390/healthcare12010092_

Round 1

Reviewer 1 Report

Comments and Suggestions for Authors

Respected authors

You and your colleagues investigated the Role of a Local Community Exercise Rehabilitation Center Through the Establishment of a Senior Living Lab, focusing on Patients with Musculoskeletal Disorders. Your manuscript is of interest but needs some revisions as follows.

-  It would be best if you benefited from a native editor. There are several punctuation and grammatical errors in the manuscript. 

- Abstract, please add more details regarding methods including the sampling method, interview type, the way you synthesize the results, and the name of the software.

- Line 17, this statement is not good enough: "The results revealed three main conclusions". Please re-write and use category or theme instead of conclusions. 

- Introduction, please add more details regarding the function of the Local Community Exercise Rehabilitation Center and  Senior Living Lab. 

- Please add your interview guide and your question as a supplementary file. 

- I am not convinced with your details of the synthesis method. What method exactly did you use to synthesize the results? It is not clear to me. Please add the method of synthesis and its steps. 

- After Section 3., please add a brief and then mention the results. This method is not transparent to readers.

Cheers

Comments on the Quality of English Language

It would be best if you benefited from a native editor. There are several punctuation and grammatical errors in the manuscript. 

Thanks

Author Response

Thank you very much for your helpful comments. We have revised the text following your recommendations and have highlighted all of our changes in yellow.  Please see the attachment.

Reviewer 2 Report

Comments and Suggestions for Authors

The manuscript – "Examining the Role of a Local Community Exercise Rehabilitation Center Through the Establishment of a Senior Living Lab: A Focus on Patients with Musculoskeletal Disorder" – is well-structured, clearly written, and methodologically sound, demonstrating a worthy effort by the authors. However, there are some issues that, in my opinion, need to be addressed before publication. Specific Comments:

1. Avoid using biased language like “elder”, “elderly”, “senior”, or “the aged", as they evoke negative stereotypes. Instead, more neutral expressions are preferred, such as “older adults".

2. In lines 65-66, the authors need to explicitly mention that the National Life Sports Survey data used in the study is from South Korea. This clarification is essential as the preceding paragraphs refer generally, and this specific context needs to be clearly stated.

3. The manuscript introduces the use of the "ICT-based Exercise Rehabilitation Service Platform" in the last paragraph of the introduction. In section 3.2.2 they show the results on older adults’ viewpoint on this. However, to provide a comprehensive background, the authors should consider discussing the prevalence and ease of use of ICT by older adults in Korea. This information could be incorporated from external sources such as this article (https://doi.org/10.3390/healthcare9091151) to enhance the context and understanding.

4. On page 7, the authors mention that “Specialized knowledge of musculoskeletal diseases in seniors is necessary, and the process of exchanging emotions is crucial [10]. Furthermore, the space in which seniors engage in physical activity goes beyond a simple physical environment, offering opportunities for various social relation-ships, role expansion, and establishing a social identity in old age”. This aspect should be further highlighted and discussed in the concluding section to underscore its significance in the overall impact of the community exercise rehabilitation center.

5. The authors avoid terms like "excellent" (line 499) and expressions of "hope" (line 510). Instead, the authors should focus on presenting findings objectively and, when discussing future directions, frame them in terms of necessity or the potential for further research (for instance, instead of "we hope that", say that, in the future, policy research ought to be conducted to).

Overall recommendation: the manuscript exhibits promising qualities but requires revisions to address the specified issues. The suggested adjustments will, in my opinion, ensure the suitability of the manuscript for publication in Healthcare.

Comments on the Quality of English Language

Overall, the text is clear and well written.

Author Response

(The authors gave the same response as above.)

Reviewer 3 Report

Comments and Suggestions for Authors

Examining the Role of a Local Community Exercise Rehabilitation Center Through the Establishment of a Senior Living Lab: A Focus on Patients with Musculoskeletal Disorder

Dear authors, thank you very much for the effort put into this work. However, there are certain weaknesses that need improvement before your publication proposal.

**Title:**

 Please specify the methodology used in the title.

**Abstract:**

In the abstract, there are two objectives that confuse the reader. Please clarify the study's objective and the methodology employed to achieve it.

**Introduction:**

- The objective stated in the introduction does not align with those in the abstract. Please unify them.

**Methods:**

- A minimum, initial, etc., sample size is indicated. However, in qualitative analyses, it is not necessary to calculate a sample size; participants should be included until data saturation is reached. Please specify how this was done.

- In section 2.1.1, clearly specify the inclusion and exclusion criteria used in the study. They are unclear due to the tedious wording.

- Indicate whether software was used for category coding and data extraction.

**Discussion:**

- Discussion involves comparing findings with those of similar studies. In this discussion, no comparisons are made as only a single bibliographic reference is included. Additionally, categorical and assertive statements are made, which are not suitable for a quantitative analysis, especially in a qualitative study with a sample size of 10 people. Please correct, rewrite, and conduct a proper discussion.

Author Response

(The authors gave the same response as above.)

Round 2

Reviewer 2 Report

Comments and Suggestions for Authors

The authors have addressed most of my comments. Apart from the need for some minor proofreading, the manuscript may in my opinion be published as it is.

Comments on the Quality of English Language

minor proofreading is needed.

Reviewer 3 Report

Comments and Suggestions for Authors

Thank you very much to the authors. They have resolved the issues raised.